# Mucolipidoses Overview: Past, Present, and Future

**DOI:** 10.3390/ijms21186812

**Published:** 2020-09-17

**Authors:** Shaukat A. Khan, Saori C. Tomatsu

**Affiliations:** 1Nemours/Alfred I. duPont Hospital for Children, Wilmington, DE 19803, USA; 2Department of Biological Sciences, University of Delaware, Newark, DE 19716, USA; stomatsu@udel.edu

**Keywords:** I-cell disease, inclusion body, lysosome enzyme transport, lysosomal storage disorders, mannose 6-phosphate, glycosaminoglycans

## Abstract

Mucolipidosis II and III (ML II/III) are caused by a deficiency of uridine-diphosphate *N*-acetylglucosamine: lysosomal-enzyme-*N*-acetylglucosamine-1-phosphotransferase (GlcNAc-1-phosphotransferase, EC2.7.8.17), which tags lysosomal enzymes with a mannose 6-phosphate (M6P) marker for transport to the lysosome. The process is performed by a sequential two-step process: first, GlcNAc-1-phosphotransferase catalyzes the transfer of GlcNAc-1-phosphate to the selected mannose residues on lysosomal enzymes in the cis-Golgi network. The second step removes GlcNAc from lysosomal enzymes by *N*-acetylglucosamine-1-phosphodiester α-*N*-acetylglucosaminidase (uncovering enzyme) and exposes the mannose 6-phosphate (M6P) residues in the trans-Golgi network, in which the enzymes are targeted to the lysosomes by M6Preceptors. A deficiency of GlcNAc-1-phosphotransferase causes the hypersecretion of lysosomal enzymes out of cells, resulting in a shortage of multiple lysosomal enzymes within lysosomes. Due to a lack of GlcNAc-1-phosphotransferase, the accumulation of cholesterol, phospholipids, glycosaminoglycans (GAGs), and other undegraded substrates occurs in the lysosomes. Clinically, ML II and ML III exhibit quite similar manifestations to mucopolysaccharidoses (MPSs), including specific skeletal deformities known as dysostosis multiplex and gingival hyperplasia. The life expectancy is less than 10 years in the severe type, and there is no definitive treatment for this disease. In this review, we have described the updated diagnosis and therapy on ML II/III.

## 1. Introduction

Mucolipidoses (MLs) are classified as a lysosomal storage diseases (LSDs) because of their involvement in increased storage materials in the lysosomes. Around 50 LSDs have been identified so far, and the incidence is approximately 1 in every 7000 births [1]. Patients with MLs are born with a genetic defect in which their bodies either do not produce enough enzymes or, in some instances, produce ineffective forms of enzymes, resulting in the accumulation of storage materials in the cells of various tissues in the body and successive damage of organs [2]. In patients with MLs, the molecules accumulate in the brain, visceral organs, and muscle tissue as well as in the bone, causing mental retardation, skeletal deformities, and poor function of vital organs such as the liver, spleen, heart, and lungs. There are four types of ML which are classified according to the enzyme(s) that is deficient or mutated: sialidosis (ML I), ML type II, initially called “inclusion cell disease or I-cell disease [3]”, now known as ML II alpha/beta (α/β) [4], ML type III (previously known as pseudo-Hurler polydystrophy, [5]), later as ML IIIA and ML IIIC, and now known as ML III α/β and ML III gamma (γ), respectively [4], and ML IV. In this review, we will focus on ML II α/β and ML III α/β in more detail and will refer them as ML II and ML III, respectively. Sialidosis is caused by the deficiency of alpha-*N*-acetyl neuraminidase due to mutations in the neuraminidase 1 gene (NEU1) resulting in the accumulation of sialylated glycoconjugates [6]. ML II is a severe form of ML in which children usually have an enlargement of certain organs, such as the liver or spleen, and sometimes even the heart. Affected children often fail to grow and develop in the first months of life. Delays in the development of their motor skills are usually more pronounced than delays in their cognitive skills. Children with ML II eventually develop a corneal clouding and, because of their lack of growth, develop short-trunk dwarfism. ML II patients exhibit clinical phenotypes at the prenatal or neonatal stage. Postnatal growth is reduced significantly within 1–2 years. ML II patients have several severe skeletal abnormalities, including craniosynostosis, osteopenia, neonatal hyperparathyroidism, rickets, thoracic deformity, kyphosis, deformed long tubular bones, hip dysplasia, clubfeet, and contractures in all large joints. Skeletal radiographs reveal signs of dysostosis multiplex that includes metacarpal pointing, bullet-shaped phalanx, oar shaped ribs, and iliac flaring. The progressive deformity of vertebral bodies presents anterior beaking or wedging and later results in kyphoscoliosis. Prenatal manifestations can include bone dysplasia with shortened and curved bones and/or bone fractures [7,8,9,10,11,12,13]. Figure 1A represents a full-body image of a 13-month-old patient with I-cell disease. The patient has a unique clinical appearance. Figure 1B shows the vertebral side image of a 9-month-old boy with I-cell disease with characteristic features of skeletal dysplasia. Peripheral blood lymphocytes of I-cell disease show abnormal vacuoles (Figure 1C).

The facies of ML patients are wizened with a bulbous nose and gingival hyperplasia [14]. Characteristic appearance includes a prominent forehead, puffy eyelids, epicanthus, flat nasal bridge, anteverted nostrils, gingival hyperplasia, and macroglossia. Gingival hyperplasia is characteristic of ML, which is associated with a cathepsin L deficiency [15]. Chondrocytes in ML are enlarged and filled with numerous vacuoles [16]. ML II patients exhibit a progressive failure to thrive, developmental delay, heart valve thickness, and calcification, and suffer from frequent recurrent upper respiratory infections. Sleep-disordered breathing is a well-recognized manifestation in patients with ML. Indeed, a flat face, a depressed nasal bridge, and the progressive deposit of metabolic substrate in the epiglottis, larynx, trachea, and base of the tongue contribute to upper and lower airway narrowing and obstructive sleep apnea [17,18,19]. Progressive mucosal thickening narrows the airways, and the stiffening of the thoracic cage contributes to respiratory insufficiency, making it the most common cause of death within the first year of life in these patients [20,21,22,23,24]. ML III (ML III α/β and ML III γ) is a milder form of ML II with a later onset of clinical signs and slower progression, enabling survival into adulthood. Joint stiffness, carpal tunnel syndrome, pain in hips, shoulders, hands, and/or ankles, waddling gait, as well as spinal deformities are common features of ML III, leading to clinical diagnosis in childhood [5,10]. Craniofacial dysmorphism, growth retardation, organomegaly, and cardiorespiratory problems are often absent or appear to be less pronounced than in ML II [25,26,27,28,29,30,31]. These young patients are often plagued by recurrent respiratory tract infections, including pneumonia, otitis media, and bronchitis. Children with ML II generally die before their seventh year of life, often as a result of congestive heart failure or recurrent respiratory tract infections [3,5]. ML II and ML III are caused by a deficiency of enzyme uridine-diphosphate *N*-acetylglucosamine: lysosomal-enzyme-*N*-acetylglucosamine-1-phosphotransferase (GlcNAc-1-phosphotransferase, EC2.7.8.17), which tags lysosomal enzymes with a mannose 6-phosphate (M6P) marker for transport to the lysosome. The process is accomplished by a sequence of two steps: first, GlcNAc-1-phosphotransferase catalyzes the transfer of GlcNAc-1-phosphate to selected mannose residues on lysosomal enzymes in the cis-Golgi network [32,33,34]. In the second step, the masking GlcNAc is removed by *N*-acetylglucosamine-1-phosphodiester α-*N*-acetylglucosaminidase (uncovering enzyme), exposing the M6P residues in the trans-Golgi network [35,36,37,38]. Figure 2 represents the stepwise modification of the *N*-linked carbohydrate chain on lysosomal hydrolases.

M6P residues on lysosomal enzymes enable M6P receptors to recognize lysosomal enzymes, followed by intracellular vesicular trafficking to lysosomes [39]. Figure 3 shows the cell function from a healthy control and an ML patient. GlcNAc-1-phosphotransferase is an α_2_β_2_γ_2_ heterohexameric enzyme [40]. The transmembrane α and β subunits are encoded by *N*-acetylglucosamine-1-phosphate transferase subunits α and β (GNPTAB) [41,42]. The enzymatic inactive transmembrane precursor protein [41,43], which is cleaved by the site-1 protease, releases catalytically active α and β subunits [44]. *N*-acetylglucosamine-1-phosphate transferase subunit γ (GNPTG) encodes the soluble γ subunit of the GNPT complex and has been shown to facilitate the recognition process [45] (Figure 4). The first cloning of human GNPTG was performed by Raas-Rothschild et al. in 2000 [45]. In 2005, Tiede et al. cloned human GNPTAB [41]. After the molecular cloning of the causative genes, ML II and ML III patients were subjected to gene analysis, and more than 175 different mutations have been identified in *GNPTAB* and *GNPTG* until now (The Human Gene Mutation Database: HGMD).

## 2. Diagnosis of ML II/III Patients

### 2.1. Clinical Diagnosis

ML II and ML III patients show clinical features similar to other lysosomal storage disorders such as mucopolysaccharidoses (MPSs); therefore, the direct measurement of UDP-GlcNAc-1-phosphotransferase (responsible for ML II and III) is ideal to differentiate ML from MPSs. However, this assay is complicated and not available in many countries [12]. In addition, enzyme assays might have some limitations in terms of interpretation. Therefore, molecular genetics and a biochemical investigation is clinically very useful in the diagnosis of ML II/III patients. A biochemical analysis using a single-chain antibody fragment that recognizes M6P has been developed [46,47] to differentiate ML patients from MPS patients. Defects in GlcNAc-1-phosphotransferase result in the missorting of lysosomal enzymes, which can be easily detected by the measurement of lysosomal enzyme activities in plasma, dried blood, and media from cultured fibroblasts or amniocytes [8,28,48,49,50]. It is important to note that ML II and ML III cannot be distinguished from each other based on lysosomal enzyme activities or M6P-containing proteins. In addition, there are no obvious clinical differences between ML III α/β and ML III γ. Therefore, a sequencing of GNPTAB and GNPTG is necessary to confirm ML II, ML III α/β, or ML III γ [29,51].

### 2.2. Genetic Diagnosis

Mutations in GNPTAB result in severe and attenuated forms of ML II and III (α/β). However, the least severe form of ML, ML III γ, is caused by mutations in the GNPTG gene. The α and β subunits contain three identifiable domains—the Stealth domain, Notch repeats, and DNA methyltransferase-associated protein (DMAP) interaction domain [52]. Qian et al. have reported 31 missense mutations in GNPTAB—15 mutations in the Stealth domain (48%), 5 mutations in Notch repeats (16%), 6 mutations in the spacer region (19%), 2 mutations in the DMAP domain, and 3 mutations were located in cytoplasmic tails [52]. The most important finding of this study was to define the missense mutation and its impact on the GlcNAc-1-phosphotransferase enzyme, such as the impaired exit from the ER and loss of catalytic activity due to mutations in the Stealth domain. This group also reported that the K732N patient mutation in the DMAP interaction domain resulted in impaired binding and decreased phosphorylation of lysosomal acid hydrolases [53]. They also reported that the K4Q and S15Y mutations in the α subunit decreased the retention of active phosphotransferase enzyme in the Golgi complex, and mislocalized into lysosomes or the ECM [54]. These findings were very important to unfold the mutations with the functional domains of α/β of the phosphotransferase enzyme. Until now, 258 different mutations in *GNPTAB* have been reported and summarized by Velho et al. [10]. Forms of mutation include frameshift mutations (39%), missense mutations (26%), nonsense mutations (23%), splice defects (9%), and deletions/duplications/insertions/deletion-insertions (3%) [10]. The GNPTAB gene contains 21 exons, and the majority of mutations (25%) are located on exon 13. These mutations facilitate the genetic diagnosis of ML. Genotype–phenotype correlations are reported in some mutations [55,56]. In general, nonsense or frameshift mutations tend to cause a severe phenotype(s), and missense mutations are involved in an attenuated phenotype(s), but this depends on each mutation. Genetic analyses suggest that the regional specificity of mutations resulted from the founder effect. For example, an extremely high rate of specific mutation c.3503_3504delTC in *GNPTAB* was found in the Saguenay-Lac-Saint-Jean region (Quebec, Canada). This mutation was introduced to Canada by immigrants from France in the 17th century [57]. The second most prevalent mutation, c.3565C > T, was identified in 43 individuals and appeared most frequently among populations in Asia but is also present in patients from Australia, Germany, Ireland, Israel, and the United States. Furthermore, 16 of 19 patients carrying the variant c.1090C > T were from China, whereas 10 of 12 patients with the variant c.1120T > C (p.R1189X) were from Japan, indicating the geographical clustering of specific mutations. In the case of Japanese ML patients, specific mutations accounted for large percentages of allele frequencies and showed a genotype–phenotype correlation [56]. For example, the allele frequency of the c.3565C > T is 41.25%, and a homozygote with this mutation shows a severe ML II phenotype. This mutation is also reported in Korean patients [58]. As for *GNPTG*, the number of patients is less than those of *GNPTAB* mutations. The GNPTG gene contains 11 exons, and at least 50 different mutations have been reported so far [10]. Most variants are intronic mutations, and the majority (45%) of them were present in intron 8. Until now, no mutation has been identified in exons 2 and 10. Velho et al. [10] have well summarized GNPTAB and GNPTG variants and pathogenicity. A specific disease caused by a defective uncovering enzyme has not been reported so far, but a recent report shows *N*-acetylglucosamine-1-phosphodiester alpha-*N*-acetylglucosaminidase (*NAGPA*) gene, which encodes an uncovering enzyme that is also associated with nonsyndromic stuttering [59].

### 2.3. Biochemical Diagnosis

ML patients may be diagnosed by the combined pattern of lysosomal enzyme activities in plasma and other biological fluids. In 1971, Wiesmann et al. reported a 90-fold increase in arylsulfatase A activity, 30- to 40-fold increase in arylsulfatase B, and a 20-fold increase in *N*-acetyl-to-β-galactosaminidase in the plasma of ML II patients, compared to those in normal controls [60]. They also reported that arylsulfatase A activity and *N*-acetyl-β-galactosaminidase activity in cerebrospinal fluid (CSF) were elevated seven times and three times, respectively. In the patient’s urine, a five-fold increase in the activity of arylsulfatase A was found [60]. In 1972, Leroy et al. reported six patients of I-cell disease with a marked decrease in multiple acid hydrolases in cultured skin fibroblasts compared to normal controls [61]. The acid hydrolases that were decreased include β-galactosidase (2% of normal), β-glucosaminidase (8% of normal), β-glucuronidase (7% of normal), α-galactosidase (10% of normal), and arylsulfatase A (5% of normal). However, β-glucosidase and acid phosphatase levels were not affected [61]. Uronic acid was determined by a uronic acid carbazole reaction (colorimetric method) [62]. Hexosamine was determined by the Boas method (colorimetric method) [63]. The content of total mucopolysaccharides in I-cell fibroblasts was not significantly different from that of normal fibroblasts. However, it is critical to measure glycosaminoglycan (GAG) levels by an advanced method such as liquid chromatography-tandem mass spectrometry (LC-MS/MS) to confirm this finding. In 1981, Reitman et al. demonstrated that the fibroblasts from patients with I-cell disease and pseudo-Hurler polydystrophy were severely deficient in *N*-acetyl-glucosaminyl phosphotransferase activity, thus identifying the biochemical basis for these diseases [64]. In 1983, Whelan et al. reported deficient activity of arylsulphatase A and B and hexosaminidase A and B in three infants’ cultured skin fibroblasts [65]. In addition, these acid hydrolases were increased markedly in plasma and in the culture medium of the skin fibroblasts. Over time, several researchers reported an increase in acid hydrolase activity in serum/plasma and a decreased activity in the fibroblasts of ML patients, which are listed below. In 1987, Tamés et al. reported an increase in the activities of multiple lysosomal enzymes in serum and a decrease in the fibroblasts of two ML patients [66]. In 2017, Yang et al. reported an increase in plasma lysosomal enzyme levels in seven patients from six families [12]. These enzymes included arylsulfatase A (25-fold), α-*N*-acetylglucosaminidase (5~26-fold), and β-hexosaminidase (7~24-fold) [12]. However, none of these enzymes’ activities increased in leukocytes. In 2017, Singh et al. reported a 22-month-old boy with ML II having MPS-like features with a significant increase in hydrolases enzymes in the plasma [67]. The levels of arylsulfatase A were 1432.2 (range: 15.5–160 nmol/h/mL); β-glucuronidase—124,999 (range: 134.5–2054 nmol/h/mL); α-L-iduronidase—1325.0 (range: 32.0–105.5 nmol/h/mg); α-iduronate-2 sulfatase—4700.0 (range: 600–1616 nmol/4 h/mg) [67].

### 2.4. Storage Materials

Historically, in 1972, Leroy et al. described that ML patients were characterized differently from MPS patients because of normal levels of urinary GAGs [68]. In 1979, Coppa et al. used column chromatography to measure urinary and tissue GAG in ML II patients. Several determinations of the daily urinary GAG excretion of the patient were found to be within the normal range. No qualitative or quantitative differences could be shown between GAGs extracted from normal and I-cell disease tissues [69]. It is possible that the method is not sensitive enough to measure these GAGs as compared to LC-MS/MS (see below). However, in 2015, Tomatsu et al. showed that ML patients had increased levels of GAGs in serum/plasma, dried blood spots, and urine specimens, which include heparan sulfate (HS), dermatan sulfate (DS), keratan sulfate (KS), and chondroitin sulfate (CS) by using advanced methods for GAG detection. These GAGs were detected by an enzyme-linked immunosorbent assay (ELISA) [68,70,71], and liquid chromatography-tandem mass spectrometry (LC-MS/MS) [69,72,73,74]. Langereis et al. also confirmed this finding by using a multiplex assay to identify all three GAGs (HS, DS, and KS) in ML II and ML III patients [72]. Otomo et al. reported an analysis of three ML II patients’ fibroblasts and found about 2-fold accumulation of phospholipids and cholesterol compared with normal cells [75]. The GAG analysis was not made in this study. The skin fibroblasts of ML patients showed an abnormal phase-dense structure in the cytoplasm, which is considered due to swollen lysosomes filled with undegraded substrates. Phospholipids are quantified by a phospholipids C-test.

In 2007, Kawashima et al. reported inclusion bodies in the lysosomes of ML II patient fibroblasts that included GM2 ganglioside, oligosaccharides, and various kinds of glycoconjugates having sialic acidα2-3galactose, galactoseβ1-4N-acetylglucosamine, and mannose residues [76]. In 2019, Yokoi et al. reported inclusion bodies in the B cells of three unrelated ML II patients, which had an accumulation of HLA class II molecules [77]. In contrast, CD4+ T cells, CD8+ T cells, natural killer cells, monocytes, or neutrophils did not contain the inclusions. These results suggest a potential role for *N*-acetylglucosamine-1-phosphotransferase in immune functions. Furthermore, the fact that only B cells contain the inclusions provides a novel diagnostic aid for the diagnosis of I-cell disease [77]. Lymphocytes include T cells, B cells, and NK cells. T cells are involved in cell-mediated immunity, whereas B cells are primarily responsible for humoral immunity. Due to lack of *N*-acetylglucosamine-1-phosphotransferase in B cells, inclusion bodies are immune-related (HLA).

## 3. Pathophysiology of ML

ML and most types of MPS share characteristic bone deformities, involving common mechanisms for these diseases. However, the degree of neuronal pathology or bone deformities, degree of disease progression, and level of GAG accumulation vary between diseases. Several lysosomal glycosidases and sulfatases enzymes are involved in a step by step degradation of GAGs, which are deficient in ML lysosomes, due to the impaired transport of these enzymes. In contrast, MPS is caused by the mutation of a single glycosidase. The relationship between GAG accumulation and systemic bone deformities remains to be determined. Since fibroblast growth factors (FGFs) are heparin-binding proteins, the extracellular accumulation of HS may affect FGFs or other signaling and possibly be involved in the bone phenotype [78,79]. In addition to GAG accumulation, the altered trafficking of lysosomal enzymes in ML may also contribute to bone phenotypes. Kollmann et al. reported that progressive bone loss in ML II is due to the presence of dysfunctional osteoblasts combined with excessive osteoclastogenesis [80]. They further underscore the importance of a deep skeletal phenotyping approach for other lysosomal diseases in which bone loss is a prominent feature. Another report using a different ML II murine model [81] revealed that cathepsin K and tartrate-resistant acid phosphatase (TRAP), two hydrolases which are essential for bone resorption, were hypersecreted to the bone resorption lacuna in ML II. Although it remains unclear whether osteoclasts are upregulated in function or number (or both), the data give an explanation for the osteopenia seen in ML patients. An accumulation of free cholesterol within lysosomes in skin fibroblasts is observed in ML, which is visualized by a fluorescent probe, Filipin [82]. Lysosomal-free cholesterol is exported to the outside by the lysosomal soluble protein NPC2 and transmembrane protein NPC1 [83]. NPC2 is transported to the lysosome by M6P-dependent pathway [84], which is impaired in ML. Lysosomal cholesterol accumulation possibly increases the lysosomal pH [75], which may further impair the export of luminal cholesterol from lysosomes [85]. The conversion of cholesteryl esters into free cholesterol is catalyzed by acid cholesteryl ester hydrolase (lysosomal acid lipase: LAL) in lysosomes. LAL is also transported by the M6P-dependent pathway, but ML fibroblasts still maintains ~18% LAL activity of normal cells [86], which suggests that M6P dependency for lysosomal enzyme trafficking differs between enzymes.

ML II is characterized by progressive neurodegeneration. Kollmann et al. developed a Gnptab-defective mouse by a single base insertion corresponding to a mutation detected in a patient with mucolipidosis II [41] to explore brain pathology in this mouse model [87]. These mice showed progressive neurodegeneration due to the loss of M6P on lysosomal enzymes. The analysis of storage material in the brain of these “knock-in” mice led to the identification of a distinct group of lysosomal proteins such as α-L-fucosidase, β-hexosaminidase, α-mannosidase and Niemann–Pick C2 protein. The accumulation of the distinct enzymes listed above is completely dependent on the M6P on lysosomal enzymes, unlike cathepsin D or B, which can use the M6P-independent pathway to transport the lysosomes of brain cells. Niemann–Pick C2 protein maintains lysosomal and autophagic functions and prevents neurodegeneration, which does not happen in the case of ML II. Some studies reported normal or increased lysosomal enzymes activities in human ML II liver autopsies [88,89,90], which is due to the M6P-independent pathway. Moreover, the uptake of circulating lysosomal enzymes into the liver of Gnptab knock-in mice by various carbohydrate-specific receptors may explain the normal or increased activities of several lysosomal enzymes [91]. Idol et al. reported neurologic consequences in ML II and III mouse models, resulting in the depletion of acid hydrolases in mesenchymal-derived cells [92]. In both cases, total brain extracts have a normal or near normal activity of many acid hydrolases, reflecting the M6P-independent lysosomal targeting pathways. The behavioral deficits occur in both models, with a greater severity in the ML II mice. The ML II mice undergo progressive neurodegeneration with neuronal loss, astrocytosis, microgliosis, and Purkinje cell depletion, whereas ML III only has a mild impact. Paton et al. [93] performed a similar study and developed a ML II mouse model that fully recapitulates the human pathology, showing growth retardation, skeletal and facial abnormalities, increased circulating lysosomal enzymatic activities, intracellular lysosomal storage, and a reduced life span [93]. They described progressive neurodegeneration in the cerebellum with a severe loss of Purkinje cells causing ataxic gait, and importantly, behavioral deficits, including impaired motor function and psychomotor retardation.

## 4. Animal and Zebrafish Models of ML

### 4.1. Animal Models

#### 4.1.1. Feline Model

In 1996, Hubler et al. reported the first animal model for ML in a cat that was short-haired with abnormal facial features and an abnormal gait [94,95]. This cat showed typical features of MPS except in levels of urine glycosaminoglycans. A set of lysosomal enzyme activities was low in fibroblasts and higher in blood plasma. Moreover, cultured fibroblasts contained numerous inclusions within the cytoplasm, and GlcNAc-1-phosphotransferase activity was deficient in leukocytes and cultured fibroblasts. A molecular analysis of this model is described below.

This model was clinicopathologically characterized by Mazrier et al. [95]. They described three lysosomal hydrolases’—α-mannosidase, β-glucuronidase, and α-fucosidase—activity in plasma and fibroblasts. The enzymes in this cat model had a 4.7-, 13.8-, and 4-fold higher plasma activity, respectively, than that of their normal littermates. In addition, the hydrolase activity in affected cat fibroblasts was reduced by 12-, 1.8-, and 9-fold, respectively, compared to that of a normal cat. The ML II cat fibroblasts also exhibited inclusion bodies [95]. In 2018, Wang et al. reported a GNPTAB nonsense variant associated with an ML II cat [96]. All affected cats were homozygous for a single base substitution (c.2644C  >  T) in exon 13 of *GNPTAB*. This variant results in a premature stop codon (p.Gln882*), which predicts severe truncation and the complete dysfunction of the GNPTAB enzyme. They described the activities of six lysosomal enzymes, which were markedly increased in the serum of the three affected kittens (α-l-iduronidase, 7-fold; arylsulfatase B, 19-fold; β-glucuronidase, 24-fold; α-d-mannosidase, 10-fold; α-d-fucosidase, 16-fold; *N*-acetyl-β-d-glucosaminidase, 7-fold), compared to that of the ten age-matched, healthy, unrelated control kittens.

#### 4.1.2. Mouse Models

Several mouse models of ML were produced by different groups using genetic engineering techniques. The first mouse model was generated by the gene trap method [97]. A mouse embryonic stem (ES) cell line carrying a mutation within *Gnptab* was obtained from a library of gene-trapped ES cell clones, which was confirmed to have an insertion of the gene trapping retroviral vector in intron 1 of *Gnptab*, causing a disruption of the full-length *Gnptab* transcript. This gene trap mouse shows growth retardation, retinal degeneration, and vacuolation in the secretory cells of exocrine glands. However, this mouse model lacks inclusions in fibroblasts and shows a normal life span. Although osteoclasts show defects in secretory lysosomes, the bone deformity is not evident.

Another mouse model was generated by introducing a human *GNPTAB* mutation (c.3145insC) into the murine *Gnptab* gene (c.3082insC) [80,87]. This mutation causes the premature termination of *Gnptab*, and no M6P-containing protein was detected in the fibroblast or brain [46]. This knock-in mouse well recapitulates human ML II phenotypes, including elevated lysosomal enzyme activities in plasma, progressive neurodegeneration, severe skeletal abnormalities, and a shortened life span. With this mouse, impaired B cell function and humoral immunity were shown [98]. Paton et al. developed a mouse model termed “Nymphe” which was identified from a phenotype-driven screen of the *N*-ethyl-*N*-nitrosourea (ENU) mutagenized mice [93]. This Nymphe mouse shows growth retardation, skeletal and facial abnormalities, increased circulating lysosomal enzymatic activities, intracellular lysosomal storage, progressive neurodegeneration, and a reduced life span, which recapitulates human pathology. The biochemical studies showed that several lysosomal enzyme activities such as β-mannosidase (2.8-fold), β-glucuronidase (4-fold), β-galactosidase (8-fold), β-hexosaminidase (12-fold), and α-mannosidase (28-fold), were markedly increased in this mouse model compared to normal controls [93]. However, the activities of α-galactosidase and β-glucocerebrosidase were normal in the serum of these mice [93]. Inclusion bodies (aggregates of polysaccharides) were present in mouse fibroblasts and secretory and connective tissues, which were analyzed under light microscopy imaging. Glycolipid (glycosphingolipids and sphingomyelin) and cholesterol deposits were observed in the brain [93]. The cholesterol was detected by filipin staining and periodic acid-Schiff staining, performed for the detection of glycolipids. A GAG analysis was not conducted for this mouse model. Chondrocytes in this mouse model included many vacuoles filled with undegraded substrates, the aggregates of polysaccharides [93].

Another mouse model was generated by the targeted disruption of *Gnptab* locus at exon 12 and exon 20 by homologous recombination with the neomycin resistance gene [99]. In this mouse, a decreased bone mineral density and skeletal abnormalities—such as spinal kyphosis, a reduced body length, and severe growth retardation—are prominent. Cultured fibroblasts contain numerous phase-dense inclusions, and lysosomal enzyme activities in culture media are markedly increased, consistent with observations in ML II patients.

The targeted deletion of exons 4–11 of *Gnptg*, the γ subunit of GlcNAc-1-phosphotransferase, was expected to provide a mouse model for ML III γ [100]. A biochemical analysis of this mouse reveals only a partial loss of mannose 6-phosphorylation, which suggests that the γ subunit enhances the mannose 6-phosphorylation of lysosomal enzymes catalyzed by α and β subunits. This mouse lacks cartilage defects and retinal degeneration, grows normally, and shows a normal life span, although predominant lesions are found in the secretory epithelial cells of exocrine glands similar to those seen in the first *Gnptab* gene trap mice [101]. This model exhibits storage vesicles in the fibroblasts but did not analyze the accumulated GAGs or lipids.

A mouse model, deficient ofithe uncovering enzyme that involves the second step of mannose 6-phosphorylation of lysosomal enzymes was generated by the insertional mutagenesis of the *Nagpa* gene [102]. This mouse is viable, grows normally, and lacks detectable histologic abnormalities, although the activities of plasma lysosomal enzymes increase mildly. M6P residues of the secreted lysosomal enzymes from this mouse were still covered by GlcNAc due to a lack of uncovering enzyme activity. The situation of these “covered” lysosomal enzymes prevents efficient recognition by M6P receptors and sorting to lysosomes, but they are sufficient for not generating tissue abnormalities seen in GlcNAc-1-phosphotransferase deficiency.

### 4.2. Zebrafish Models

Flanagan-Steet et al. developed the GlcNAc-1-phosphotransferase deficient zebrafish model utilizing the morpholino knockdown method [103]. Morphant embryos exhibit multiple phenotypes, such as an abnormal craniofacial development, impaired motility, and defects in the otic vesicle structure. Due to loss of phosphotransferase activity, these embryos showed an increased expression of type II collagen and transcription factor Sox9, resulting in the alteration of chondrocyte differentiation and homeostasis of the extracellular matrix, which is a common pathology of ML patients. This group further demonstrated the increased levels of cathepsins K, L and S and MMP-13 during the developmental stages of an ML II embryo [104]. The increased activity of cathepsin K caused a defect in the cartilage; however, the inhibition of cathepsin K reduced several proteases and partially corrected the craniofacial phenotypes of these ML II embryos [104]. These results indicate the role of cathepsin K in the cartilage pathogenesis of ML II patients. Flanagan-Steet et al. also described the selective mannose phosphorylation using GNPTG-deficient zebrafish [105]. These embryos lack the gross morphological or craniofacial phenotypes compared to GNPTAB-deficient morphant embryos. They reported that the loss of γ subunit reduces mannose phosphorylation on a subset glycosidase; however, cathepsin proteases are unaffected. This is due to the fact that cathepsin proteases are normally expressed in GNPTG-deficient embryos, the chondrocyte morphology is normal, and the expression of collagen type II is also normal. However, the basis for the selective effects of the γ subunit on glycosidase mannose phosphorylation in the zebrafish system is unclear. This group also demonstrated that the abnormal bones and cartilage in ML II are due to the cathepsin-mediated upregulation of transforming growth factor β (TGF-β) signals and reduction in bone morphogenetic protein (BMP) signals during chondrogenesis in zebrafish [106]. Furthermore, TGF-β mediates the regulation of cathepsin K during normal and pathogenic development in a zebrafish model [107].

Qian et al. used the zebrafish model to study Notch domain mutations. They described five mutations—all block formations of different disulfide linkages [52]. Three Notch 1 mutations have no impact on the catalytic activity of GlcNAc-1-phosphotransferase but caused an impairment in the recognition of lysosomal hydrolases. In contrast, Notch 2 mutants significantly reduced the activity of GlcNAc-1-phosphotransferase due to the partial impairment of the endoplasmic reticulum (ER) exit. However, it is not clear whether Notch 2 mutations have a role in lysosomal hydrolase recognition.

## 5. Therapy and Management of ML Patients

There is no definitive treatment for ML. Patients should be cared for with supportive and symptomatic management to prevent life-threatening events and maintain the quality of their life. In this section, we have described supportive therapies and other possibilities for therapies.

### 5.1. Supportive Therapy

Supportive therapy includes many types of interventions, such as physiotherapeutic intervention, surgical intervention, and bisphosphonate therapy. Physiotherapeutic interventions accommodate patients’ specific needs and conditions, requiring careful planning and procedures. “Low impact” aqua therapy pertaining to joint and tendon strain in ML III patients is seen to be well received and tolerated [108]. As the ML disease progresses, the variety in bone pain increases, promoting the management of the pain by surgical procedures. In cases involving pain in the knee and hip, knee replacements and bilateral hip replacements have been successful. When there is a disruption in ventricular function due to valvular dysfunction, heart valve replacement is highly considered. To prevent bacterial endocarditis, premedicated antibiotics are recommended. In addition to premedication, there are also concerns about airway management for patients with ML, and anesthesia should be well planned [108,109]. With regards to bisphosphonate therapy, patients with a significant skeletal disease and a decrease in bone mineral density (z score < −2.5) have been managed with this form of therapy [30,108]. ML patients suffer from severe osteopenia, the form hyper-resorption of the bone. Bisphosphonate counteracts this by inhibiting osteoclasts and ultimately preventing bone resorption. Clinical trials of bisphosphonate therapy on ML II patients indicated reduced bone pain and an improved quality of life, without any significant changes both biochemically and histologically [109].

### 5.2. Hematopoietic Stem Cell Transplantation (HSCT)

Hematopoietic Stem Cell Transplantation (HSCT) has demonstrated its use as a possible treatment for lysosomal storage disorders. HSCT provides “donor-derived hematopoietic cells that produce lysosomal enzymes with the M6P moiety, allowing for intracellular uptake with appropriate trafficking to the lysosomal for substrate degradation” [110]. Lund et al. reported the outcomes of HSCT on children with ML II [110]. In this study, 22 patients with ML II underwent HSCT, and the results revealed the death of 12 of the patients were related to complications due to the disease itself, rather than the transplant procedure. The neurologic follow-up post-transplantation, however, showed mixed results in the effectiveness of HSCT [110]. This led to an insufficient conclusion in the ability of HSCT to improve the clinical outcomes. Similar results were evident in another long-term case study with a 12-month ML II female patient. The patient had received allogeneic bone marrow transplantation (BMT) at 19 months for the prevention of cardiorespiratory complications and the continuation of intellectual development. At the age of 7 years, the patient showed no progression in cardiac or respiratory issues and had been achieving neurodevelopmental milestones at a slow rate. Despite the limitations, BMT did appear to point towards some beneficial effects on growth and intellectual development, although the mechanism remained unclear [111]. Naumchik et al. described the role of hematopoietic cell transplants in several glycoproteins including ML (sialidosis, mucolipidosis II, mucolipidosis III) [112]. With the current understandings of HSCT, further modifications to the therapy are encouraged as well as the use of the therapy along with other different treatments.

### 5.3. Future Therapies

Future therapies for ML call for different possibilities. With uncertainty in the effectiveness of many of the therapies mentioned, a combination of the various therapies is currently utilized for treatment.

#### 5.3.1. Enzyme Replacement Therapy (ERT)

Enzyme Replacement Therapy (ERT) replaces defective or deficient enzymes in the body of patients with lysosomal storage diseases. ERT is currently available for several LSDs (MPS I, MPS II, MPS IVA, MPS VII, and Pompe disease). For MPS I patients, the effects of ERT were limited in skeletal dysplasia and cognitive function, while an association between the duration of ERT and growth of children was seen among those with MPS II. Although relatively positive outcomes are evident in visceral organs, the impact of ERT to the bones and brain was restricted [113]. Despite many studies on the effects of ERT on other MPSs, there have not yet been any clinical or animal studies of ML on the effect of ERT, as the GlcNAc-1-phosphotransferase enzyme is a membrane protein. An in vitro study by Hickman et al. has demonstrated that ML cells can internalize exogenous mannose 6-phosphorylated lysosomal enzymes to lysosomes, and that ML cells treated with a mixture of several lysosomal enzymes can improve cellular functions [75,114]. The challenge in ML is that many lysosomal enzymes are deficient in the cells because of a lack of M6P residue. It remains an unmet challenge to produce multiple recombinant mannose 6-phosphorylated lysosomal enzymes and to treat patients with the enzyme mixture.

#### 5.3.2. Gene Therapy

A recent study with gene therapy involved a mouse model treated with a recombinant adeno-associated viral vector (AAV) of GNPTAB. In comparison to the untreated knock out mice, the treated mice displayed a significant increase in both bone mineral density and content. The results also presented a decrease in IL-6 in articular cartilage for the treated mice. Although the exact mechanisms remain obscure, such data propose the hypothesis that via the inhibition of IL-6 production, bone loss can be decreased [99,109]. More studies in ML mice models are required in order to get the maximum outcome from therapy, which has not been achieved at present.

#### 5.3.3. Pharmacological Chaperon

Another possible treatment is the idea of pharmacological chaperone proteins that can bind to mutated enzymes to allow better stability or intracellular trafficking. Candidate chaperones that can improve GlcNAc-I-phosphotransferase or regulate lysosomal enzyme trafficking are yet to be discovered through a screening of chemicals or approved drugs [109,115]. Another therapeutic option for ML may be a soy isoflavone, genistein, based on the concept of substrate reduction. Genistein is a specific inhibitor of tyrosine-specific protein kinase activity of the epidermal growth factor (EGF) receptor [116], and reduces GAG synthesis [117]. There are several reports describing genistein’s effects on MPS. A treatment of up to 160 mg/kg/day on an MPS IIIB mouse decreased the total GAG level in the liver, spleen, and urine, but the brain pathology was not changed [118]. Clinical trials on MPS III patients indicate that genistein at a dose of 10 mg/kg was not effective. Clinical trials on MPS II patients revealed an improvement of connective tissue elasticity and a range of joint motion at a dose of 5 mg/kg/day [119]. Meanwhile, genistein increased GAG in MPS I chondrocytes and fibroblasts and decreased body length and femur length in the MPS I mouse model [120,121]. Therefore, there is no clear evidence for the effect of genistein in MPS, especially for the brain. There is no clinical trial of genistein for ML. When ML skin fibroblasts were treated with genistein, HS accumulation was reduced. However, cellular growth was also inhibited by genistein dose-dependently [122], probably because general cellular metabolism was inhibited due to a blockade of signaling from growth factors by genistein. A further accumulation of clinical cases treated with genistein and the elucidation of molecular mechanisms are necessary for drawing conclusions.

#### 5.3.4. Antisense Oligonucleotides

Recently, Matos et al. have developed an antisense oligonucleotide-based exon skipping strategy to treat ML II [123]. A deletion of a dinucleotide (c.3503_3504del) on exon 19 of GNTAB makes truncated GlcNAc-1-phosphotransferase, resulting in a complete loss of enzyme activity. However, skipping exon 19 results in GNTAB having 56 amino acids less, which does not result in a complete loss of activity and thus provides an important future therapy for this disease.

## 6. Conclusions

Several research groups and physicians have provided a greater understanding for ML as well as the M6P-dependent transportation of lysosomal enzymes since ML has been identified. However, the perspective of ML pathophysiology or M6P-independent pathways remains to be elucidated. Further accumulation of clinical and basic experiences is necessary for the complete understanding and cure of ML.

## Figures and Tables

**Figure 1 ijms-21-06812-f001:**
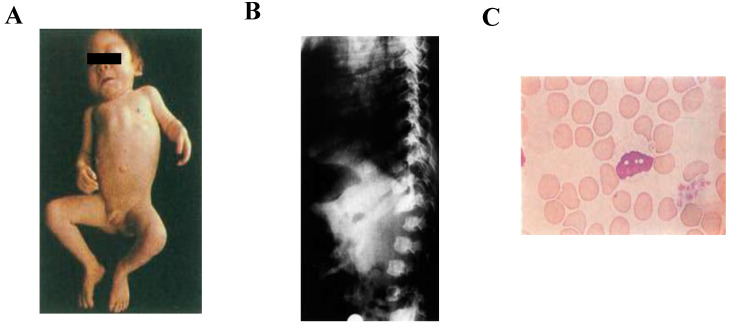
Clinical feature of I-cell disease (kindly provided by Dr. Tadao Orii). We have signed the inform consent (**A**). A full-body image of the patient with I-Cell disease (a 13-month-old boy). The patient has a distinct coarse face, short neck, umbilical hernia, thick skin, and rigidity of joints. (**B**). The vertebral side image of a patient with I-cell disease (a 9-month-old boy). Anteroposterior diameter of the vertebral body and ossification of the anterior upper border of the lumbar vertebral body are reduced. Dysfunction is observed and hump-back of the vertebral L2 body. (**C**). Peripheral blood lymphocytes (at 40× magnification) of I-cell disease (mucolipidosis II) (May-Giemsa staining). The cytoplasm is filled with numerous vacuoles.

**Figure 2 ijms-21-06812-f002:**
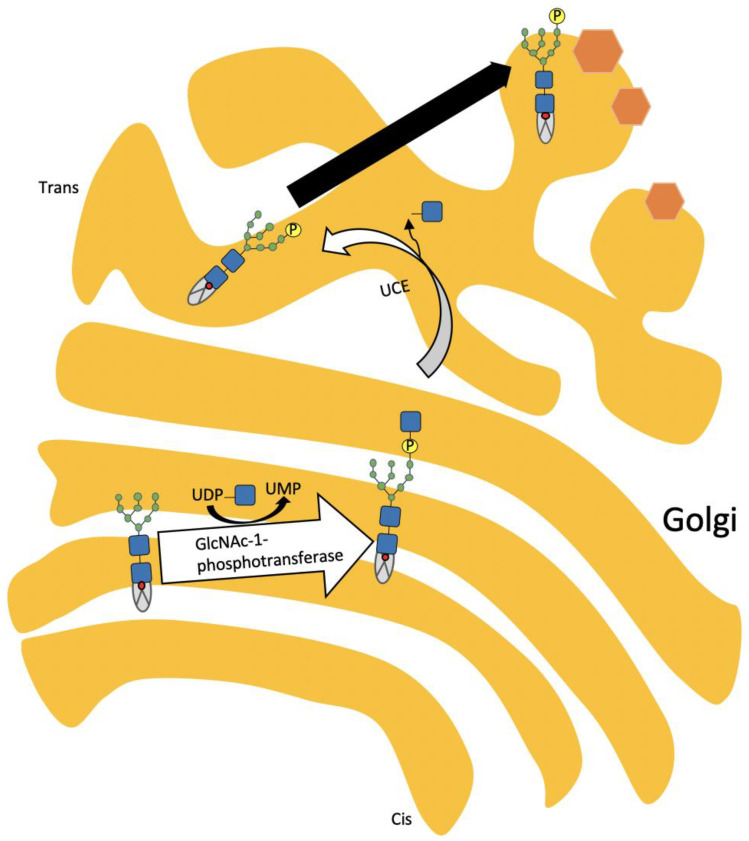
Stepwise modification of *N*-linked carbohydrate chain on lysosomal hydrolases. GlcNAc-1-phosphotransferase phosphorylates selected mannose residue on lysosomal enzymes, followed by the removal of GlcNAc from lysosomal enzymes by the uncovering enzyme. The phosphorylated enzymes bind with the mannose 6-phosphate (M6P) receptor and are internalized to the endosome and finally to the lysosomes.

**Figure 3 ijms-21-06812-f003:**
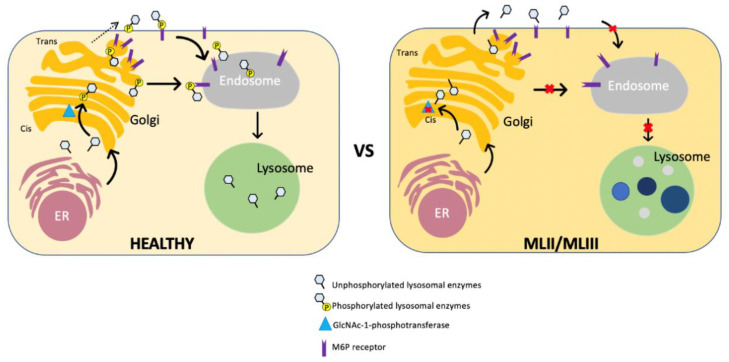
Comparison of cell function between healthy control and ML II/III patients, adapted from Velho et al. [10]. The healthy cells phosphorylate lysosomal hydrolases by GlcNAc-1-phosphotransferase in the cis-Golgi network. The uncovering enzyme removes the masking of GlcNAc to expose M6P residues in the trans-Golgi network. The M6P-exposed enzymes are taken by endosomes with the M6P receptor and are delivered to lysosomes. In ML II/III cells, defective GlcNAc-1-phosphotransferase is unable to phosphorylate lysosomal hydrolases in the cis-Golgi network, which are targeted to the ECM, resulting in accumulation of storage materials in the lysosomes.

**Figure 4 ijms-21-06812-f004:**
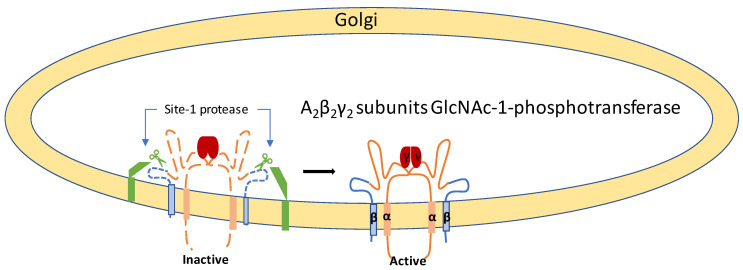
Structure of GlcNAc-1-phosphotransferase in Golgi. Adapted from Velho et al. (2019) [10]. GlcNAc-1-phosphotransferase comprises the α_2_β_2_γ_2_ heterohexameric transmembrane protein. The site-1 protease cleaves inactive precursor protein and releases catalytically active α and β subunits.

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
