# Peer review of "Mucolipidoses Overview: Past, Present, and Future"

_ijms, 2020, doi:10.3390/ijms21186812_

Round 1

Reviewer 1 Report

Thank you for making it easy for me to review this manuscript "Mucolipidoses overview: past, present, and future" 

It's a good, comprehensive review of mucolipidosis. In my opinion the review is quite comprehensive, however the future section remains underdeveloped in particular that concerning gene therapy and haematopoietic precursor transplantation which should be expanded and discussed for each of the subtypes of mucolipidosis.

Matos L, Vilela R, Rocha M, et al. Development of an Antisense Oligonucleotide-Mediated Exon Skipping Therapeutic Strategy for Mucolipidosis II: Validation at RNA Level. Hum Gene Ther. 2020;31(13-14):775-783.

Naumchik BM, Gupta A, Flanagan-Steet H, et al. The Role of Hematopoietic Cell Transplant in the Glycoprotein Diseases. Cells. 2020;9(6):1411.

Author Response

Comments and Suggestions for Authors (reviewer 1)

Thank you for making it easy for me to review this manuscript "Mucolipidoses overview: past, present, and future" 

It's a good, comprehensive review of mucolipidosis. In my opinion the review is quite comprehensive, however the future section remains underdeveloped in particular that concerning gene therapy and haematopoietic precursor transplantation which should be expanded and discussed for each of the subtypes of mucolipidosis.

Matos L, Vilela R, Rocha M, et al. Development of an Antisense Oligonucleotide-Mediated Exon Skipping Therapeutic Strategy for Mucolipidosis II: Validation at RNA Level. Hum Gene Ther. 2020;31(13-14):775-783.

Naumchik BM, Gupta A, Flanagan-Steet H, et al. The Role of Hematopoietic Cell Transplant in the Glycoprotein Diseases. Cells. 2020;9(6):1411.

Response: We appreciate the comments of the reviewer to improve our manuscript. We have included “Development of an antisense oligonucleotide-mediated therapeutic strategy for mucolipidosis II” under future therapy, which is the latest future therapy that we miss in this review. Another reference, “The role of hematopoietic cell transplant in the Glycoprotein Disease. Cells. 2020;9(6):1411” is a latest report for treatment of glycoprotein diseases by HCT. In this article, the authors covered HCT treatment in glycoproteins, including ML (sialidosis, mucolipidosis II, mucolipidosis III). In our review paper, we described HCT for ML II in Section 4.3. Hematopoietic Stem Cell Transplantation (HSCT). We have added this reference in this section.

Reviewer 2 Report

With recent advances in our understanding of ML disease and the GNPTAB-related disorders, a review on this topic is timely for the field.  The authors attempt to summarize a great deal of historical and recent information in this area but the overall focus and organization make it difficult for the reader to follow.  There are numerous corrections and modifications that are needed throughout the review in order for it to serve as a comprehensive and authoritative overview of this topic.  Perhaps most discouraging are certain omissions that are also highlighted below (including research on animal models and a large body of information regarding the functional significance of different conserved domains in the GNPTAB enzyme).  Furthermore, there is only a fleeting mention of M6P-independent sorting pathways and the discussion on its relevance to pathophysiology is absent.  An important note on nomenclature: the authors should carefully examine the 2008 paper (Molecular order in mucolipidosis II and III nomenclature. Cathey SS, Kudo M, Tiede S, Raas-Rothschild A, Braulke T, Beck M, Taylor HA, Canfield WM, Leroy JG, Neufeld EF, McKusick VA.Am J Med Genet A. 2008 Feb 15;146A(4):512-3.) and update the corresponding nomenclature in their review.  Corrections and necessary modifications are listed below:

Introduction:

1) Some clinical insights on MLIII should be provided to complement the information provided on MLII (see below for the suggestion to reorganize this section.)  

Figures:

1) The schematic in Figure 1 is unnecessary and confusing and should be removed.  in the second part, the authors should use conventional symbols for the glycan representation (mannose: green circles; GlcNAc: blue square).  Also, the position of the phosphate on the glycan is not accurate as there is trimming of the terminal glycans prior to the action of GlcNAc-1-phosphotransferase.  The use of blue for the Golgi and lysosomal enzyme makes it difficult to see.

2) Figure 2 is too crudely presented to be informative.  I don't see the GlcNAc-1-phosphotransferase enzyme at all in the healthy panel nor is the secretion of hydrolases depicted in the MLII panel.  The authors are strongly encouraged to redesign this panel into a more effective summary of the cellular consequences associated with MLII/MLIII.

3) Figure 3 is similar to that found in the following paper: Site-1 protease and lysosomal homeostasis. Velho RV, De Pace R, Klünder S, Di Lorenzo G, Schweizer M, Braulke T, Pohl S.Biochim Biophys Acta Mol Cell Res. 2017 Nov;1864(11 Pt B):2162-2168).  If not changed, this source should be credited as "figure adapted from".

4) The X-ray in Figure 4 is low resolution and the reviewer had a difficult time appreciating the pathology that is being described.

Clinical diagnosis: as a section title, this is somewhat misleading as it really describes the phenotypic distinctions between MLII and MLIII.  This section should be moved above to where the clinical features are first introduced.  Then the section on how the diagnosis of ML or GNPTAB-related disorders is achieved, with a highlight on both molecular testing but also analysis of plasma glycosidase activity to distinguish between ML and other LSDs.

Genetic diagnosis: the geographic information here is not as important to highlight in my view.  Instead the authors should summarize the outstanding studies from the Kornfeld lab that describe the functional significance of different domains in GNPTAB as defined by mutations within these domains.  This work not only provided fundamental insight into the function of the GNPTAB gene product but made a major contribution towards understanding the genotype/phenotype correlations in ML disease.  Unique phenotypic presentations such as those found in patients with the K4Q were subsequently defined.  

Storage materials: line 197, Leroy is misspelled.

Pathophysiology of ML: this section is far too brief in light of the wealth of information now available from recent studies in numerous animal models.  There appears to be no mention of effects on the nervous system or brain and I found the discussion about what is happening in different tissues to be inadequate.  Framing this section with the information learned from the different animal models would be far more effective.

Animal Models: this section should appear before the discussion of therapies and tied more closely to the section on pathophysiology.  While the work on the feline and murine models is presented, the extensive work on a zebrafish model for MLII is completely omitted.  This is a glaring oversight as the studies in zebrafish span more than 10 years, have produced numerous high impact publications, and resulted in important new insight into the role of secreted cathepsin proteases in the ML pathophysiology and the regulation of growth factor signaling.

Therapy and management of ML: this section could easily be condensed there is only data regarding the efficacy of HSCT (not favorable).  The other proposed therapies lack any experimental proof to date.  I would shorten this section considerably and instead add in more detail on both animal models and pathophysiology.

Author Response

Comments and Suggestions for Authors (reviewer 2)

With recent advances in our understanding of ML disease and the GNPTAB-related disorders, a review on this topic is timely for the field.  The authors attempt to summarize a great deal of historical and recent information in this area but the overall focus and organization make it difficult for the reader to follow.  There are numerous corrections and modifications that are needed throughout the review in order for it to serve as a comprehensive and authoritative overview of this topic.  Perhaps most discouraging are certain omissions that are also highlighted below (including research on animal models and a large body of information regarding the functional significance of different conserved domains in the GNPTAB enzyme).  Furthermore, there is only a fleeting mention of M6P-independent sorting pathways and the discussion on its relevance to pathophysiology is absent.  An important note on nomenclature: the authors should carefully examine the 2008 paper (Molecular order in mucolipidosis II and III nomenclature. Cathey SS, Kudo M, Tiede S, Raas-Rothschild A, Braulke T, Beck M, Taylor HA, Canfield WM, Leroy JG, Neufeld EF, McKusick VA.Am J Med Genet A. 2008 Feb 15;146A(4):512-3.) and update the corresponding nomenclature in their review.  Corrections and necessary modifications are listed below:

Response: We appreciate the comments of the reviewer to improve our manuscript. We have included ML II and ML III nomenclature, as suggested by the reviewer.

Introduction:

1) Some clinical insights on MLIII should be provided to complement the information provided on MLII (see below for the suggestion to reorganize this section).  

Figures:

1) The schematic in Figure 1 is unnecessary and confusing and should be removed.  In the second part, the authors should use conventional symbols for the glycan representation (mannose: green circles; GlcNAc: blue square).  Also, the position of the phosphate on the glycan is not accurate as there is trimming of the terminal glycans prior to the action of GlcNAc-1-phosphotransferase.  The use of blue for the Golgi and lysosomal enzyme makes it difficult to see.

Response: We have removed the first part of figure 1 (now figure 2). We have changed symbols of glycan representation to conventional symbols. We have fixed the position of the phosphate on the glycan prior to the action of GlcNAc-1-phosphotransferase. We have also changed the color of Golgi and lysosomal enzymes, which are easy to see now.

2) Figure 2 is too crudely presented to be informative.  I don't see the GlcNAc-1-phosphotransferase enzyme at all in the healthy panel nor is the secretion of hydrolases depicted in the MLII panel.  The authors are strongly encouraged to redesign this panel into a more effective summary of the cellular consequences associated with MLII/MLIII.

Response:: We have made GlcNAc-phosphotransferase enzyme visible in figure 2 (now figure 3), and added other details to make the figure informative. We have also included the text that the figure was “adapted from Velho et al.”

3) Figure 3 is similar to that found in the following paper: Site-1 protease and lysosomal homeostasis. Velho RV, De Pace R, Klünder S, Di Lorenzo G, Schweizer M, Braulke T, Pohl S.Biochim Biophys Acta Mol Cell Res. 2017 Nov;1864(11 Pt B):2162-2168).  If not changed, this source should be credited as "figure adapted from".

Response: We have referred figure 3 (now figure 4) as “Adapted from Velho et al.”

4) The X-ray in Figure 4 is low resolution, and the reviewer had a difficult time appreciating the pathology that is being described.

Response: We understand the concern of reviewer. Unfortunately, we do not have an alternative picture.

Clinical diagnosis: as a section title, this is somewhat misleading as it really describes the phenotypic distinctions between MLII and MLIII.  This section should be moved above to where the clinical features are first introduced.  Then the section on how the diagnosis of ML or GNPTAB-related disorders is achieved, with a highlight on both molecular testing but also analysis of plasma glycosidase activity to distinguish between ML and other LSDs.

Response: We have moved the text at the beginning of the manuscript where the phenotype of MLII/III has described for the first time. We have described molecular testing and plasma hydrolases activity to distinguish between ML and other LDSs.

Genetic diagnosis: the geographic information here is not as important to highlight in my view.  Instead, the authors should summarize the outstanding studies from the Kornfeld lab that describe the functional significance of different domains in GNPTAB as defined by mutations within these domains.  This work not only provided fundamental insight into the function of the GNPTAB gene product but made a major contribution towards understanding the genotype/phenotype correlations in ML disease.  Unique phenotypic presentations such as those found in patients with the K4Q were subsequently defined.  

Response: We have revised the text, removed the geographic information, and summarized the outstanding studies from the Kornfeld group.

Storage materials: line 197, Leroy is misspelled.

Response: We have corrected the spelling of Leroy.

Pathophysiology of ML: this section is far too brief in light of the wealth of information now available from recent studies in numerous animal models.  There appears to be no mention of effects on the nervous system or brain and I found the discussion about what is happening in different tissues to be inadequate.  Framing this section with the information learned from the different animal models would be far more effective.

Response: We have added ML II/III mice models and an impact on nervous system. Additional pathophysiology of ML is included in this section. We have also included a detailed mannose-6-phosphate independent lysosomal targeting pathway.

Animal Models: this section should appear before the discussion of therapies and tied more closely to the section on pathophysiology.  While the work on the feline and murine models is presented, the extensive work on a zebrafish model for MLII is completely omitted.  This is a glaring oversight as the studies in zebrafish span more than 10 years, have produced numerous high impact publications, and resulted in important new insight into the role of secreted cathepsin proteases in the ML pathophysiology and the regulation of growth factor signaling.

Response: We have improved this section by including zebrafish models used in ML to show the role of cathepsin proteases in ML pathophysiology. We have also included regulation of TGF-bsignaling.

Therapy and management of ML: this section could easily be condensed there is only data regarding the efficacy of HSCT (not favorable).  The other proposed therapies lack any experimental proof to date.  I would shorten this section considerably and instead add in more detail on both animal models and pathophysiology.

Response: We agree with the reviewer and have made changes in this section to make the text shorter.

Round 2

Reviewer 2 Report

My requested changes were met with reasonable satisfaction.  The term "cathepsins K" instead of "cathepsin K" is used several times and should be corrected.  

Author Response

We have changed cathepsins K to cathepsin K in the text.